# Gatekeepers in the health financing scheme: Assessment of knowledge, attitude, practices, and participation of Malaysian private general practitioners in the PeKa B40 scheme

Mohammad Husni Jamal[1,2], Aznida Firzah Abdul Aziz[3], Azimatun Noor Aizuddin[4,5]*, Syed Mohamed Aljunid[5,6]

1 University of Cyberjaya, Cyberjaya, Malaysia, 2 Academy of Family Physicians of Malaysia, Kuala Lumpur, Malaysia, 3 Department of Family Medicine, Faculty of Medicine, Universiti Kebangsaan Malaysia, Bangi, Malaysia, 4 Department of Public Health Medicine, Faculty of Medicine, Universiti Kebangsaan Malaysia, Bangi, Malaysia, 5 International Centre for Casemix and Clinical Coding, Hospital Canselor Tuanku Muhriz, Universiti Kebangsaan Malaysia Medical Centre, Kuala Lumpur, Malaysia, 6 International Medical University, Kuala Lumpur, Malaysia

* azimatunnoor@ppukm.ukm.edu.my

**Data Availability Statement:** Data cannot be shared publicly due to legal and ethical reasons. Data used in this research is available on

## Abstract

This is cross-sectional research done to assess the readiness of the private Malaysian general practitioners (GPs) for the implementation of the national health financing scheme. The study focused on their levels of knowledge and attitudes towards the types of health financing scheme, gatekeeper roles in the health financing scheme, and their participation in the PeKa B40 scheme. Their acceptance and level of participation in the national health financing scheme (NHFS) were also assessed. A set of self-designed and pre-tested questionnaires focusing on the aforementioned objectives were mailed to the respondents. The selection of respondents was done by stratified random sampling of the GPs in all 14 Malaysian states at both urban and rural levels. Out of a calculated number of 362 GPs targeted, 296 responses were received which represented a response rate of 81.7%. The respondents had a mean age of 50.7 years 165 (55.75%) were males and 131 (44.3%) were females. The rural respondents totalled 158 (53.4%) as compared to those from urban 138 (46.6%) areas. The outcomes observed were that GPs with PeKa B40 provider status, positive attitude towards health financing schemes, gatekeeper roles, and PeKa B40, were strongly associated with their acceptance and level of participation in the NHFS. The GPs possessed a positive attitude and were generally ready to participate in the NHFS, but the lower scores in knowledge levels would require definite education and training plans to further enhance their readiness. More incentives should be given to GPs to enrol as PeKa B40 providers. The results of this study should be strongly considered by the government in the efforts to engage the Malaysian private GPs in the forthcoming NHFS. Most importantly, the role of GPs as gatekeepers needed to be implemented, and the PeKa B40 scheme be greatly improved.

reasonable request to UKM Research Ethics Committee (contact via sepukm@ukm.edu.my) for researchers who meet the criteria for access to restricted/ confidential data.

**Funding:** Yes. The study was funded by the Faculty of Medicine, Universiti Kebangsaan Malaysia FF-2020-403 under supervisor's name Azimatun Noor Aizuddin for PhD student's research. the funders had no role in study design, data collection and analysis, decision to publish, or preparation of the manuscript.

**Competing interests:** The authors have declared that no competing interests exist.

## Introduction

A national health financing scheme (NHFS) is designed to protect a country's population concerning the coverage of the costs of their health care. This entity is usually created by national legislation such as an Act of Parliament. The stakeholders may come from either the public or private sector, or both. The funding mechanisms may differ, depending on each country's needs. The structure of a health financing scheme can comprise government schemes such as general taxation, social health insurance, compulsory private insurance, compulsory medical savings account, voluntary health insurance schemes, non-profit institutions finance schemes such as foreign aid, enterprise finance schemes in which firms can opt for employees coverage, household out-of-pocket expenses and finally a "rest of the world (ROW)" for non-residents in which the participants may be either compulsory or voluntary. Additionally, "ROW" benefits entitlements are decided by foreign entities who are the fund providers and the fund pooling vary across the programmes. The number and interaction of the components depend on the needs of each particular country [1]. Ensuring efficiency, equity, and equality in creating healthcare systems is the topmost priority [2].

It had been reported that by the 1990s, Malaysia had successfully achieved universal health coverage with regard to basic healthcare [3]. Despite this achievement, however, Malaysia lacked a national-coordinated health financing mechanism for its healthcare service delivery. Malaysia was without any mandatory, payroll-based scheme such as social or national health insurance, apart from two small Social Security schemes i.e. Employees Provident Fund (EPF) and the Social Security Organisation (SOCSO). Currently the system was mainly funded by general taxation in the public sector. The increasing public demand for better quality care, added to issues of changing demographics and disease burdens, compounded the Malaysian health system under further pressure [4].

In the latest Health White Paper report tabled in the Malaysian Parliament 15th June 2023 recently, the latest data up to 2020 are revealed. The Malaysian government spent only a Total Health Expenditure (THE) equivalent to 4.1% of the Gross Domestic Product (GDP) in both the public and private sectors. This figure is much lower than the average of 7.4% of GDP for other upper-middle-income (UMIC) countries and 8.8% for high-income countries (HIC). Additionally, Malaysia's public sector contribution of current health expenditure is only 2.2% of GDP. This figure is also similarly low compared to the UMIC average of 4.4% of GDP and the HIC average of 6.4% of GDP. The duality of public and private healthcare services has caused a highly imbalanced distribution of services, infrastructure, equipment, manpower and other resources across both sectors. The outcomes register an overstretched public healthcare facilities and human resources compared to those in the private sector. The number of public sector clinics comprise 28% of the total primary healthcare facilities but have to manage almost 64% of outpatient visits. At the same time, a high level of out-of-pocket payments (OOP) is noted, especially in the private sector. As a funding source, OOP is inefficient and inequitable as it raises financial risk for households, lowers risk pooling and supports avoidance of care. The current OOP expenses is the second largest component of current health expenditure (CHE), which amounts to 34.2% of CHE in Malaysia. In other UMIC countries such as Thailand and Turkiye, the average OOP health expenditure stands at 8.7% and 16.9% respectively. Another issue is the relatively high proportion of its public health resource allocation on hospital care as compared to clinic care. Between 1997 to 2020, public expenditure on hospital care has shown steeper increments in comparison to public expenditure on clinic care [5].

Achieving universal health coverage requires an effective structuring and management of health financing. Overall, there is a need for good governance and leadership so as to reduce inefficiencies and wastages of public funds and resources. Equitable financing coverage should

encompass the three areas of depth, breadth and proportion of health costs that will be covered. Coverage of the poor and the whole care spectrum from preventive and promotive to curative and rehabilitative measures need consideration. In Malaysia the duality of public and private health services may potentially weaken the case for universal coverage due to certain disparities. Notably, the private healthcare sector is subject to the control of market mechanisms and competition. The financing sources include out-of-pocket, private medical insurance, and individual savings (Employees Provident Fund), Additional prepayment financing mechanisms have been considered for the past two decades in the form of a social health insurance structure to supplement the financing of escalating healthcare costs but until today, this has not materialised [6].

Efforts to establish a national health financing system in Malaysia has been in the making for the past three decades. The first study is the National Healthcare Financing Study by Westinghouse in 1984–1985. A series of studies followed, and the last proposed healthcare reform is known as 1CARE in 2009 by the Ministry of Health of Malaysia (MOH). However, despite the 1CARE proposal being a comprehensive one, it is lacking in details and evidence, allowing misinterpretation, and became the centre of controversy among stakeholders. This proposal is subsequently deferred to this day [7].

There is growing evidence that political economy factors are central to the adoption or rejection of health financing reforms, there are three areas which contributed to this argument: Firstly, the relative strengths of "pro" or "against" interest groups. Secondly, specifically the number and strength of formal 'veto gates' in the political decision-making process Thirdly, the 'policy feedback' effects, whereby different patterns of public opinion and mobilisation of interest groups are created by various political regimes. In the context of Malaysia, it is observed that the most important factor is 'policy feedback' effects which obstructs changes to Malaysia's health financing system. Secondly is the interest group opposition, as the politicians view these groups as public opinion. Thirdly, institutional veto gates, by contrast, play a minimal role in preventing health financing reform in Malaysia. A major fact causing the many resistances to health reforms can be attributed to Malaysia's early success at achieving near-universal access to public sector healthcare at low cost. The main groups against reform are civil society actors (consumer groups and trade unions), professional medical associations and opposition political parties. The professional medical associations include the private sector doctors, who comprise approximately half of all doctors in Malaysia. And they are represented in the Malaysian Medical Association (MMA), the Federation of Private Medical Practitioners Associations, Malaysia (FPMPAM) and the Association of Private Hospitals Malaysia (APHM). Among clinicians, the largest and most influential in health financing reform dialogues are the estimated 7000 private general practitioners (GPs) in Malaysia. Opposition from private GPs has proven to be a major barrier to the implementation of the 1Care reform. Specialist clinics in particular strongly oppose 1Care, as they would potentially lose patients and income under a system that incorporates a family medicine 'gatekeeper'. On the other hand, others are concerned that non-specialist GPs who have fewer patients (and lower revenues) will be potentially agreeable to a capitation-based system that will increase their patient numbers. In this case, their support or opposition will critically depend on the capitation rates to be agreed with the government. 1CARE is a commendable effort to reform the health finance system. It will have been supported by the already existing general taxation with the additional introduction of social health insurance. The reform will significantly result in a better coverage and positive outcomes but unfortunately the government of the day has to bow down to opposition voices objecting to the "insurance component i.e., social health insurance". This component is unfavourable with many sectors of the population [8].

However, in recent years, MOH has expressed serious interest in the adoption of a National Health Insurance scheme. In 2019, in a town hall meeting with Malaysian general practitioners, the then MOH (2018 to 2020), Datuk Seri Dr. Dzulkefly Ahmad, proposes the Malaysian government to increase the budget allocation for health between 6% to 7% from the current 4.5% then. He also stresses a reformed health financing system that will move from a predominantly current tax-based system to one that allows the incorporation of an insurance system. In the same meeting, the government further reemphasises that the general practitioners (GPs) will play an important role as gatekeepers in the proposed scheme, to reduce unnecessary admissions to tertiary care [9]. His successor, Minister of Health Khairy Jamaluddin from August 2021, subsequently continues this initiative, recommending that social health insurance will be one of the schemes that would be incorporated [10].

Further evidence of the impending review is reaffirmed with the Ministry of Health (MOH) White Paper Summit which is held in August 2022 involving all the stakeholders. The Minister states that in order to move from "sick care" to health care and wellness, the Health White Paper will need to address the imbalance between primary health care and hospital care in terms of resource distribution, organisation, and policy focus. Evidence from many global experiences has proven that increasing emphasis on investment in primary health care will lead to better health outcomes. These includes a reduction in the overall cost of care and improved equity and access. Additionally, there is also some reduction in the adverse impact of social inequalities on health. Furthermore, the lessons of the COVID-19 pandemic substantiate those countries with strong primary healthcare systems are able to respond better and faster to the pandemic, with minimal disruption of essential services [11].

In a more recent statement, the minister reemphasizes that the focus will be on better health financing through an adequate, efficient, and equitable public sector that will ensure a sustainable health system. Additionally, this can be achieved through strengthening primary healthcare (PHC). He further remarks that the core towards achieving universal health coverage (UHC) is via PHC, which essentially provides a much wider range of multidisciplinary health services, thus assuring more comprehensive and continuous care [12].

Apart from the considerations on tax-based and social health insurance, private health insurance is another form of funding mechanism that we consider in our nationwide survey that is done. However, they are mainly premised on the ability to pay, and not on needs. Private health insurance can be classified into substitutive (to the existing statutory type), complementary (coverage for services excluded or not fully covered by the state), or supplementary (coverage for faster access and increase consumer choice).

In the Malaysian setting, the delivery of primary care is contributed to both by the public and private sectors. In the public health sector, primary care services are fully subsidised by the government. The services are provided by doctors in the health clinics or Klinik Kesihatan (KK) and university-based primary care clinics, whereas the GPs in the private sector provide primary care services for anyone who can afford to pay for services rendered. Therefore, the focus on primary care delivery should be on both sectors. Additionally, both the public and university primary care clinics are now led by vocationally-trained family medicine specialists (i.e. Master's degree qualification or its equivalent). The government is also encouraging more private general practitioners to be trained to become specialists in family medicine to streamline primary care services. A well-coordinated public-private partnership in primary care must also be instituted. For example, this partnership will enable the burden of care of chronic disease management to be shared between public and private GPs within a particular community.

If primary care is to be at the forefront of the healthcare system, increased allocation must be given to primary outpatient care, which will help reduce the strain. A major shift in the

allocation of the budget towards the primary care sector needs to be implemented. This is because various multilevel studies have proven that a good primary care system result in better health outcomes and equity as well as cost reduction [13].

Strong primary care is required to achieving universal health coverage. In many low and middle-income countries, the private primary care sector is expanding due to various challenges in the public sector. These include financial constraints, shifts in disease burden from communicable to chronic non-communicable diseases, demographic changes and political and economic instability. These result in many countries having mixed health systems that comprise of public and private providers including Malaysia. Public primary care clinics are under the purview of Ministry of Health Malaysia (MOH), while the private primary care clinics are privately owned practices managed by general practitioners. Three quarters of these GP clinics are solo practices too. In comparison, most public clinics comprise of a mix of care providers such as family medicine specialists, medical officers, nursing staff, physiotherapists, occupational therapists, and pharmacists. The public clinics are more widely distributed nationwide but the private GP clinics are mainly in urban, affluent areas. The differences in the quality and performance between both sectors need to be understood. These include organisational structure, financing sources, governance and regulations, and in provider and patient characteristics. Both public and private primary care services have their strengths and weaknesses respectively. The public sector is lagging behind in providing comprehensiveness, continuity and coordination of care. Hence, improvements required include strengthening specialist feedback to promote seamless care. Patient referrals need to be more patient centred with reduced administrative requirements. On the other hand, the primary focus in the private sector should be on incentivising the private providers to improve the delivery structure such as comprehensive care and to increase inter- and intra-professional cooperation. The emergence of new challenges such as an ageing population and the rising non-communicable disease burden must be considered. Strengthening the primary care service delivery has been shown to be one of the critical measures in overcoming these challenges. By identifying the strengths and weaknesses of each sector, the government can devise a strategy to have an effective public-private collaboration [14].

Private Primary Health Care Private medical services in Malaysia are governed by the Private Healthcare Facilities and Services Act 1998, This Act ensures increased access to health care, standardisation of quality of care, and improve provider payment. The Act also allows equitable distribution of accredited facilities managed by qualified health and allied health professionals. Private clinics greatly outnumber public clinics but were mostly concentrated in urban areas. Private clinics were preferred for treatment for acute conditions despite the comparatively high charges. However, public health care facilities were the choice for inpatient care, especially those from low-income groups. The Ministry of Health funds the public sector whilst private health care is usually financed from household out-of-pocket (OOP) expenditure, private corporations or companies, private health insurance, non-governmental organisations, and managed care organisations. The classification and assessment of healthcare facilities consider the areas of accessibility, availability, accommodation, affordability, and acceptability. The Malaysian government has succeeded in providing nearly free primary health care service to the citizens. Minimal fees for secondary and tertiary health care facilities are charged. In areas of health financing, experts have demanded the government's genuine commitment in establishing national social health insurance by integrating the appropriate regulatory measures and institutions, standardising and extending the SOCSO and EPF programmes, and achieving political and civil support. The recognition and expert management of both financial and provision components are factors required to improve primary health care in Malaysia [15].

Even more important is the role of the GPs as gatekeepers in such systems. The experiences of countries with organised health financing systems i.e., clearly defined GP gatekeeping roles, have had positive effects on the cost-effectiveness of primary care services, and subsequent lower expenditures on secondary and tertiary hospital services, despite the concerns of some dissatisfied patients who prefer direct access to specialists [16].

A gatekeeper is a focus entry point for a healthcare need. There are two sides of the argument between the patient-doctor relationship in gatekeeping. On one hand, the patient is the decision-maker and the physician should cater to his individual patients, and not the general population. The concern suggests the trust-bad relationships between patient and physician will be possibly eroded by the physician's economic position and tasks. On the other hand, proponents of gatekeeping argue that primary care physicians will be able to integrate their societal arguments in a morally acceptable way, by balancing the spheres of "values "and "rationality". Additionally, the necessity for gatekeeping is supported by needs such as that patients receive appropriate care, budget controls, and to ensure justice in care allocation. Further supporting arguments suggest that both doctors and patients must control health care costs. Otherwise, other parties may get involved which will cause more serious damage. Furthermore, societal issues need to be discussed by primary care physicians with their patients with frankness and honesty, so as to prevent the danger of losing their patients' trusts. This presents the main issue of how they needed to walk the "balancing act" i.e., to form a bridge between the logic of clinical judgment as opposed to the logics of cost-effectiveness and justice. Rational action in any sphere requires not only a state of mind, but also requires social, psychological, and material resources and skills. Effective gatekeeping will also require the capacity to acknowledge sufficient technical resources, such as specific information on pricing of procedures, possessing good diagnostic skills, the assistance of supportive technology, and having sufficient knowledge and tools such as guidelines [17].

In Malaysia, the role of the primary care doctor as a gatekeeper is already being practised in the public healthcare system. Public health clinics or KKs are manned by primary care doctors and more recently by vocationally-trained family medicine specialists. This ensures appropriate utilisation of services for patients who will subsequently be referred to secondary or tertiary services in the public hospitals. However, in the private sector there is no formal gatekeeping function by the GPs. Patients have the freedom of choice to access the private specialists directly. However, in many cases, GPs who have been appointed as panel doctors for private companies or semi-government institutions may be required to execute some form of gatekeeping by ensuring that consultations by specialists for their employees will need referrals by the GPs and subsequent approval of the management or relevant authorities of the institutions [18].

In a comparative 6-country study in the Asia-Pacific region, Australian general practitioners function as a gatekeeper to secondary care, with specialist access through GP referrals only. Whereas in Malaysia, only primary care in the public sector has a gatekeeper function. Gatekeeping has been proven to be a powerful instrument to reduce inequalities and promote integrated care. Hence, gatekeeping needs also to be introduced into the private sector [19].

The expenditure of managing GP clinics has increased tremendously. This is partly due to the changes in policies as well as the involvement of third-party administrators. It is vital that the role of GPs in primary care services in Malaysia need to be recognised by the government. GP clinics provide extended accessibility due to their longer operating hours. Additionally, the role of GPs need to be reviewed. They should be vocationally-trained, and they need to include preventive services and management of chronic diseases. The objection by GPs on third-party administrators must be seriously considered by the government as it is unfavourable to their clinic practices [20].

One particular national health financing scheme that strongly supports the general practitioner as a gatekeeper is the Australia Medicare scheme. Medicare is the insurance scheme that gives Australian citizens and permanent residents access to healthcare, including a wide range of health and hospital services at no cost or minimal cost. Medicare is a tax-based funding scheme in which Australian taxpayers contribute 2% of their taxable income to help cover costs. The coverage includes coverage of medical services by doctors, specialists and other health professionals, hospital treatment, coverage of prescription medicines as well as mental health care. Medicare covers (or refunds) the full schedule fee for GP services, 85% of the schedule fee for specialist services as well as 75% of the schedule fee for in-hospital services [21].

In a proposal to reform the Medicare scheme, there is a shift away from the fee-for-service patient rebate system. This is to accommodate the integrated care teams of GPs collaborating with practice nurses, allied health professionals and pharmacists. This is especially relevant for chronic disease management. This reform towards GP-led multi-disciplinary integrated care teams is supported by RACGP President Dr Nicole Higgins [22].

The Australian Medicare scheme is relevant for consideration in Malaysia, as it has taken into importance the role of GPs as gatekeepers as well as their stewardship in leading integrated primary care teams.

Another model that may be considered by Malaysia is the system of Universal Health Coverage (UHC) in Thailand. It consists of a hybrid financing scheme of two tax-based entities and one social health insurance scheme. The first tax-based scheme is the Civil Servant Medical Benefit Scheme (CSMBS) for government servants and families, and has a population coverage of 7%. The second tax-based Universal Coverage (UCS) package for other residents especially the poor, has the largest coverage of 75%. The social health insurance based Social Security Scheme (SSS) for the private sector employees provides for 16% of the population. The Comprehensive Benefit package covers extensively from preventive, curative, maternal and child health cares. Others include medication, rehabilitation and emergency services. Equally important is the concept of purchaser-provider split with the gatekeeper role by primary care doctors. Fundamental to the success of Thailand's UHC is support and collaboration between healthcare professionals, civil societies and politicians [23].

The gatekeeping role by GPs can have wider implications in improving the primary healthcare system delivery. This is especially so in relation involving interprofessional collaboration by GPs with other allied health professionals in primary care. For instance, there is a positive potential of the gatekeeping role by GPs to their collaboration with community pharmacists. In a qualitative study that involved 12 interviews of 5 GPs, 5 pharmacists and 2 nurses, a conceptual framework was sought to enhance interprofessional collaboration between GPs and CPs (community pharmacists) under a collaborative medication therapy management model (CMTM) in Malaysia. This model hopes to be able to showcase an effective chronic disease management at primary care level which is currently not existing in the private sector as opposed to the dispensing separation that exists in public primary care clinics. Currently, there has been a poor collaboration between GPs and CPs in the private primary care sector. It has aggravated into a conflict or business rivalry. This issue needs to be addressed and resolved since more than 41% of Malaysians seeks treatment in the private sector. The study concludes that the most feasible remuneration model for CMTM in the Malaysian setting will be either third- party payer or universal health coverage. This can be achieved by the implementation of a national health financing scheme with GPs assuming the gatekeeper role, and collaborating with the CPs according to the proposed CMTM model. The proposed model includes the defined protocols for collaborative practices, incentives for both CPs and GPs, revision of relevant legislation to grant CPs an active role in chronic care, and frequent monitoring by the

Ministry of Health. Dispensing separation will no more be a conflicting issue as both professions will be adequately renumerated in a win-win situation. Both GPs and CPs need to change their mindsets and work together. Secondly, public awareness of the advantages of the CMTM model needs to be well promoted [24].

This concept can eventually be extended in the national health financing scheme to consider GP/family physician-led integrated teams providing a cost-effective healthcare at the primary care level with the integration of more health professionals. As an early initiative to encourage the active participation of private GPs, the Malaysian government has launched the *Peduli Kesihatan* scheme (PeKA B40) by the then Honourable Minister of Health Datuk Seri Dr Dzulkefly Ahmad in 2019. The PeKa B40 is a medical scheme under the management of Protect Health Corporation, a non-profit subsidiary of the MOH. This healthcare package is introduced to cater to the 40% low-poverty sector of the population otherwise known as the B40 group. The median monthly income of this group is less than RM three thousand only. All those above 40 years old are eligible. It is a pilot programme as a public-private partnership between private GPs and primary care doctors in government health clinics. With an initial allocation of RM 100 million, the programme hopes to benefit 800,000 members of the population in the B40 group. The PeKa B40 programme covers benefits that includes screening for non-communicable diseases (NCDs) by GPs, financial support allocation for the purchase of medical devices, cancer treatment, and transportation for hospital care [25]. Although it is a form of public private primary care partnership, the focus of the study is on private Malaysian GPs because the main issues concern their lack of effective participation. The role of the GPs is limited to performing screening services only. This superficial and limited role, as well as the low remuneration rates, are the major obstacles to its acceptance by the GPs. For their services, the GPs are only paid RM 40 for the first visit and another RM 20 for the follow-up visit. These issues need to be made clear and resolved. Only then will the public private primary care partnership between the GPs and public primary care doctors be positively significant.

Recently, the then Health Minister Khairy Jamaluddin who succeeds Dr Dzulkefly Ahmad, reveals that 90% of six million Malaysians eligible for free health screenings under the Health Care Scheme of the B40 Group (PeKa B40) do not make use of the service. Only 600,000 people have tapped into the benefits of the scheme in the last three years [26].

Additionally, other possible shortcomings need to be analysed and resolved for more meaningful participation by the private GPs. The implementation of a National Health Financing Scheme for Malaysia has been a subject of much discussion, research, and recommendations for the last 30 years. In the absence of such a system, the current role of GPs as gatekeepers in Malaysia will remain informal and unstructured.

The objectives of this study are to assess the private Malaysian GPs' knowledge and attitude toward health financing schemes, gatekeeper roles, and the PeKa B40 system. In addition, it is to evaluate their participation in the impending national health finance scheme (NHFS).

## Methods and materials

### Population and sample

The study involved all private general practitioners (GPs) and their clinical practices that registered with the Medical Practice division of the Ministry of Health (2019) Malaysia as well as with the Malaysian Medical Council. This cross-sectional study was conducted from May 2021 to April 2022. The inclusion criteria considered all privately registered general practitioners, either solo or group practices with active email addresses. The exclusion criteria rejected clinics with less than two years of establishment, clinics run by secondary specialities, such as internists or paediatricians, and aesthetic clinics. The GP respondents were obtained from

members of the Malaysian Medical Association, the Malaysian Primary Care Network (MPCN), *Persatuan Doktor Islam Malaysia* (PERDIM), the Federation of Private Medical Practitioners Associations of Malaysia (FPMPAM), and the Academy of Family Physicians of Malaysia as well as other GP networks.

The Cochran method was applied for calculating the sample size using n = $Z^2$pq / $e^2$ formula with Z = 1.96 for 95% of the Confidence Interval, p = proportion of attribute under study observed previously, q = (1 −p) and e = precision (commonly used is 5%). The proportion 'p' 62.2 was selected from a study done by Blecher et al [27].

n = $1.96^2$(62.2) (0.38) / $(0.05)^2$ = 363.2. In order to have an equal balance between urban and rural clinics, the number of 362 was opted for.

The selection of sample was by Stratified Random Sampling. The sampling population was first divided into two subgroups or strata, which were either urban or rural, based on their locations in each of the 14 states. For this nationwide study, "urban" referred only to the state capitals, and any other location was classified as "rural". Hence, from the sample size of 362 clinics, it was desirable to have at least 181 urban and 181 rural clinics. Following this, each of the strata of 181 urban and rural practices was categorised according to their states, which required at least 13 practices in each urban/rural strata from each state to be selected by random sampling. The disproportionate method of random sampling was applied in each state. At least 20 practices from each urban/rural stratum were randomly selected by the use of a random number table to achieve the desired minimum of 13 practices in each stratum. The initial response from the federal societies such as the Malaysian Medical Association (MMA) and FPMPAM was poor. In order to obtain a bigger number of responses for sampling, we approached each state via different networks such as the state-level MMAs and the other formal GP groups. Sources were also obtained from other various informal GP connections. Hence, we had to adopt different means from state to state but finally we managed to obtain the required number for the stratified sampling. However, we had 66 non-responders despite several reminders. Whilst we should have attempted to recruit more participants, the difficulty faced was the survey was carried out over the period of the Covid lockdown. This was unpredicted for but we had no choice but to continue. It was very difficult to get more GP responders as the mood amongst GPs was very passive.

## Questionnaire development

The self-designed questionnaires involved the contents from the literature review. Both face and content validation of the questionnaires was done by selected experts in general/family practice and health finance systems. A pilot test of the survey questions was done by 30 private Malaysian GPs who did not participate in the actual survey.

Some examples of the articles in the literature review in relation to the survey questions were as follows:

A. Knowledge of health financing schemes

Question no 1. Tax-based finance refers to financing resources derived from government revenues

Tax-based financing referred to a health financing system in which health expenditures were derived mainly from government revenues and the accessibility to the health services was open to all citizens. More specifically, this referred to a system in which more than half of public expenditure was financed through revenues that were unlike payroll taxes such as social health insurance. This was one of the most common funding mechanisms, which had its pros and cons [28].

Question no 9: In SHI, payment models can be either single or multi-payor systems.

In a descriptive study to consider the implementation of SHI in Pakistan, the experiences of developed (e.g.Germany and Netherlands) and developing countries (e.g. Thailand, Mexico and South Africa) were studied. As for consumer choices, the Netherlands offered flexibility in the free choice of consumers. Payment models could be in the form of a single-payer or multi-payer system. The advantages of a single-payer system were ease of revenue collection, efficiency, overall cost control and subsidy coverage for the poor [29].

Question no 24: Private insurance is not the preferred choice for Universal Health Coverage

Private health insurance in an unregulated environment would lead to unethical practices. Therefore, there was a need to regulate the activities of private insurances. Public financed healthcare was the preferred choice to achieve universal health coverage via general taxation or social health insurance. In no way can private insurances achieve this ethically [30].

B. Knowledge of gatekeeper

Question no 1: A GP gatekeeper triages the patient's further access to the health system.

A gatekeeper was described as a focus entry point for a healthcare need. The function of the GPs' to meet patients' demand with their medical requirements becomes increasingly complex with the growing multiple conditions population. Shared decision-making (SDM) was now the acceptable approach to the treatment decision making process. However, variations to this ideal were observed whereby decision-making were dominantly by GPs or patients [31].

C. Knowledge of PeKa B40

Question no 5: The PEKA B40 scheme covers those from the minimum age of 40 years old.

Recipients of the Sumbangan Tunai Rahmah (STR) and their spouses aged 40 and above are automatically eligible for PeKa B40. No special registration is required to join PeKa B40 [32].

In relation to the questionnaire development, they are self-administered because they are cost-effective, easy to administer for small and large groups and self-paced. They are close-ended for nominal and ordinal variables. Likert scales are adopted. Open-ended questions are limited to some responses from PeKa B40 providers eg How many patients have you signed up since registration?

In the idea generation phase, a pilot study of the survey questions involving 30 GP respondents by convenient sampling provided no major flaws. Additionally, via the Smart PLS protocol, the questionnaires passed the convergent and discriminant validity as well as multicollinearity analysis. Hence there were no modification nor deletion of the survey questionnaires.

The final set of questionnaires consisted of five headings, including (1) demographic characteristics such as age, gender, race, nature of the practice, practice hours, years in operation, practice location, ownership, postgraduate qualification, PeKa B40 provider status (2) knowledge of health financing schemes, gatekeeper roles, PeKa B40 and sources of information S1–S4 Tables (3) attitude towards health financing schemes, gatekeeper roles and PeKa B40 S5–S7 Tables, (4) practice issues in PeKa B40 and provider/non-provider responses S8 Table and finally (5) acceptance and level of participation to the national health finance system (NHFS) S9 Table.

Knowledge of health financing schemes was assessed with a series of 24 questions which comprised six on general taxation, ten on social health insurance, and eight on private insurance; on gatekeeper roles with eight questions and on PeKa B40 with five questions. Sources of

knowledge consisted of seven items which were named the internet, mass media, social media, courses, societies, peers, and finally journals. Questions on attitude to health financing schemes numbered 15, to gatekeeper roles numbered four whilst to PeKa B40 listed as eight. There were 16 questions on practice issues in PeKa B40, which were categorised into history components (five), physical examination (six), and investigation (seven). Finally, six questions were listed to assess the level of acceptance and participation in the NHFS.

The evaluation of the questionnaires was based on the Smart PLS method of assessing the strength of the reflective measures (indicators) used in the construct based on the results of the convergent and discriminant validity. These results would establish that the construct measures were valid and reliable.

The outer model assessment evaluated the convergent validity which was achieved since the constructs achieved all the minimum values for indicator reliability ($> 0.7$), composite reliability ($> 0.7$), and average variance extracted (AVE) $> 0.5$. Cronbach's alpha was no more a preferred measure as it had been superseded by the aforementioned measures. Discriminant validity was achieved using the Fornell-Lacker criterion. The inner model assessment of collinearity on all constructs showed that all values are $< 3.3$. These implied that there were no collinearity issues in the model. Based on the above assessment overall, the model was therefore reliable and valid.

Excluding demographics, the responses to each questionnaire were graded according to the 5-point Likert scale and the scores allocated were from Strongly disagree 1 –disagree 2 –neutral 3 –agree 4 –Strongly agree 5. Neutral means neither disagree nor agree. In order to assess the final categories, scores were assessed by adapting the original Bloom's cut-off points, 80.0%–100.0% (good), 60.0%–79.0% (moderate), and <59.0%, (poor). Mean scores were also calculated.

## Data collection

Data were collected via self-administered questionnaires which were mailed to all 362 selected sample. The initial response was 319/362 participants. However, 3 respondents replied that they rejected to participate. From this balance of 316 respondents, 20 were excluded because of being less than 2 years in practice. The final response rate was 296/362 i,e 82%. This leaves a balance of 66 non-responders who failed to respond despite several reminders.

This non-response bias is most likely attributable to the repeated Covid lockdown situations when the survey was carried out. In considering the possibility of other non-response biases, we had ruled out other common causes including:

1. Poor survey design: We conducted a pilot test involving 30 GPs. We did not encounter any specific complaints on the survey questionnaires including the time duration taken to complete the survey. We also avoided structuring double-barrelled questionnaires.

2. Failed delivery: non-responders were monitored and communicated with, and some had initially experienced such surveys being sent in their spam boxes but were retrieved.

3. Data collection period: We avoided a rushed or short data collection periods/reminders by giving a reasonable duration of about 2 weeks for each respondent. Beyond this time reminders via personal telephone calls/ chat messages were given.

## Data entry and statistical analyses

The data was entered and analysed using the International Business Machine Statistical Package for Social Sciences (IBM SPSS) Statistics V.27.0 Categorical data were presented as frequency and percentages, whilst continuous data as means and standard deviation (SD).

### Ethical approval

Ethical approval was awarded by the UKM Ethics Committee letter dated August 28, 2020 (ref no UKM PPI/111/8/JEP-2020-539).

**Ethics statement (S1 File).** The respondent information sheet S2 File and informed consent forms S3 File were attached as initial pages of the questionnaires. The respondents would indicate their written consent on the attached forms and subsequently respond to the survey questionnaires. Upon completion, the information sheet and written consent forms were submitted together with the responses. The consent of each participant was ensured that it was duly filled before the responses were accepted.

## Results

### A. Descriptive statistics

**Sociodemographic characteristics.** As shown in Table 1, the mean age (±SD) of respondents was 50.7 (±12.6) years. The majority of them were males (55.7%) and Malay (56.4%). In terms of the clinic practice, most of them were solo practitioners (62.8%), operated non-24 hours practice (90.2%), and had been in operation for six to ten years (16.2%). The highest proportion was noted among Kelantanese practitioners (8.4%), and in the rural area at 53.4%. Almost half of them were the principal owners (48.6%) and 60.5% were non-PeKa B40 providers. Moreover, more than half of the respondents did not have any Family Medicine postgraduate qualifications, at 62.2%. The average duration of being a PeKa B40 provider was two years. Further clarification to the "Nature of practice "and "Ownership" are as follows: Group practices are typically divided into single-specialty and multispecialty practices. In this case, being GPs, they are of the single specialty type. The defining characteristic of single-specialty group practice is the presence of two or more GPs providing patients with one specific type of care ie primary care. "Employee" refers to GPs being employed within one of several practice models. In this case, the GP is employed according to the local labour laws. Unlike salaried doctors, GP partners own a share of their practice and draw income from business profits. As a partner, one is responsible for the running of the practice and will be expected not only to provide clinical sessions, but also to play an active role in the administrative and business side of the practice. Unlike salaried GPs, there is no set income range for GP Partners. As directors, the clinic practice is structured along corporate lines. Hence, each director owns ownership by the number of shares and the finances and working structure is guided by the Companies Act. Subsequently, apart from being working directors, they can also employ other doctors to work in their group practice.

The above categories in the study are all referring to the Malaysian private GP settings and not in the university-based settings.

The majority of respondents (60.5%) rejected the PeKa B40 scheme. Table 2 below showed the reasons for not registering as PeKa B40 providers, according to the order of importance. The most important reason cited was incomplete patient management 58.1% (agree 33.5% + strongly agree 24.6%), followed by poor remuneration 45.8% (agree 27.9% + strongly agree 17.9%), and complicated registration procedures 42.5% (agree 25.7% + 16.8%). On the lower levels were no interest 20.6% (15.6% + 5.0%) and the least important reason was not of aware of the PeKa B40 program 12.3% (6.7% + 5.6%).

The assessment of scores by Bloom's cut-off points (described earlier in the Methodology section), was observed in Table 3 below. The majority of GPs had moderate scores in all of the components under study. More specifically, as for knowledge, a substantial number possessed a good level related to gatekeeper roles (44.6%), followed by a low level on PeKa B40 (27%) but

**Table 1. The sociodemographic characteristics of general practitioners in Malaysia (N = 296).**

| Sociodemographic variables | Frequency | % | Mean (sd) |
|---|---|---|---|
| **Age (years)** | - | - | 50.7 (12.6) |
| **Gender** | | | |
| Male | 165 | 55.7 | |
| Female | 131 | 44.3 | |
| **Race** | | | |
| Malay | 167 | 56.4 | |
| Chinese | 68 | 23.0 | |
| Indian | 45 | 15.2 | |
| Sabah indigenous | 6 | 2.0 | |
| Sarawak indigenous | 2 | 0.7 | |
| Others | 8 | 2.7 | |
| **Nature of practice** | | | |
| Solo | 186 | 62.8 | |
| Group | 110 | 37.2 | |
| **Practice hours** | | | |
| 24 hours | 29 | 9.8 | |
| Non 24 hours | 267 | 90.2 | |
| **Years in operation** | | | |
| Less than 2 | 2 | 0.7 | |
| 3–5 | 36 | 12.2 | |
| 6–10 | 48 | 16.2 | |
| 11–15 | 47 | 15.9 | |
| 16–20 | 39 | 13.2 | |
| 21–25 | 41 | 13.9 | |
| 26–30 | 36 | 12.2 | |
| 31–35 | 16 | 5.4 | |
| More than 35 | 31 | 10.5 | |
| **Practice location** | | | |
| Johor | 22 | 7.4 | |
| Kuala Lumpur | 12 | 4.1 | |
| Labuan | 5 | 1.7 | |
| Putrajaya | 6 | 2.0 | |
| Kedah | 20 | 6.8 | |
| Kelantan | 25 | 8.4 | |
| Melaka | 23 | 7.8 | |
| Negeri Sembilan | 24 | 8.1 | |
| Pahang | 21 | 7.1 | |
| Penang | 15 | 5.1 | |
| Perak | 21 | 7.1 | |
| Perlis | 11 | 3.7 | |
| Sabah | 23 | 7.8 | |
| Sarawak | 24 | 8.1 | |
| Selangor | 23 | 7.8 | |
| Terengganu | 21 | 7.1 | |
| **Practice Location** | | | |
| Urban | 138 | 46.6 | |
| Rural | 158 | 53.4 | |

(*Continued*)

**Table 1.** (Continued)

| Sociodemographic variables | Frequency | % | Mean (sd) |
|---|---|---|---|
| **Ownership** | | | |
| Principal | 144 | 48.6 | |
| Partner | 56 | 18.9 | |
| Director | 37 | 12.5 | |
| Employee | 59 | 19.9 | |
| **Family Medicine postgraduate qualification** | | | |
| Diploma in FM/ Graduate Certificate in FM | 47 | 15.9 | |
| MAFP–FAFP/ FRACGP | 31 | 10.5 | |
| M Med (Fam Med) | 3 | 1.0 | |
| Others | 31 | 10.5 | |
| None | 184 | 62.2 | |
| **PEKA B40 provider** | | | |
| Yes | 117 | 39.5 | |
| No | 179 | 60.5 | |
| **Total** | **296** | **100.0** | |
| **Duration of being PeKa B40 provider (n = 117)** | - | - | 2.2 (0.9) |

even poorer levels of understanding of health financing schemes. (17.6%). A strongly positive attitude towards gatekeeper roles had the highest score (36.5%), but almost similarly poor levels for health financing schemes (16.2%) and PeKa B40 (18.2%). A strong majority had positive responses to practice matters in PeKa B40 (64.9%) which implied that the consultation format was acceptable. Finally, one-third (31.1%) of the respondents gave a very strong indication of acceptance and level of participation in NHFS, coupled with those with a moderate score of 39.9% totalling 71%, suggesting a potentially good response towards its implementation.

As shown in Table 4, knowledge of gatekeeper roles had the highest mean score, at 4.08 (0.49) as compared to the knowledge of health financing schemes and PeKa B40. The higher score indicated better knowledge among general practitioners.

Table 5 showed the mean comparison of attitude components. Attitude towards gatekeeper roles had the highest mean score, at 3.96 (0.70), indicating better attitude as compared to other attitude components.

Table 6 displayed the comparison of the means in practice issues in PeKa B40, whereby all three components attained high scores with the history element having the highest value of 4.50 (0.60).

Table 7 showed the mean score of the overall acceptance and preparedness in the participation of the National Health Financing Scheme (NHFS) was calculated below. The mean value was 3.73.

**Table 2. Reasons for not registering as PeKa B40 provider (n = 179).**

| No | Item | n (%) | | | | |
|---|---|---|---|---|---|---|
| | | **Strongly disagree** | **Disagree** | **Neutral** | **Agree** | **Strongly agree** |
| **1.** | **Incomplete patient management as it only involves screening** | **7 (3.9)** | **8 (4.5)** | **60 (33.5)** | **60 (33.5)** | **44 (24.6)** |
| **2.** | **Poor remuneration** | **10 (5.6)** | **9 (5.0)** | **78 (43.6)** | **50 (27.9)** | **32 (17.9)** |
| **3.** | **Complicated registration procedures** | **7 (3.9)** | **9 (5.0)** | **87 (48.6)** | **46 (25.7)** | **30 (16.8)** |
| 4 | Not interested | 39 (21.8) | 18 (10.1) | 85 (47.5) | 28 (15.6) | 9 (5.0) |
| 5 | Not aware of the PeKa B40 program | 68 (38.0) | 37 (20.7) | 52 (29.1) | 12 (6.7) | 10 (5.6) |

**Table 3. Assessment by overall scores according to Bloom's cut-off classification.**

| Variables | Categories | n | % |
|---|---|---|---|
| Knowledge of health financing schemes | Good | 52 | 17.6 |
| | **Moderate** | **177** | **59.8** |
| | Poor | 67 | 22.6 |
| Knowledge of gatekeeper roles | Good | 132 | 44.6 |
| | **Moderate** | **145** | **49.0** |
| | Poor | 19 | 6.4 |
| Knowledge of PeKa B40 | Good | 80 | 27.0 |
| | **Moderate** | **138** | **46.6** |
| | Poor | 78 | 26.4 |
| Attitude towards health financing schemes | Good | 48 | 16.2 |
| | **Moderate** | **187** | **63.2** |
| | Poor | 61 | 20.6 |
| Attitude towards gatekeeper roles | Good | 108 | 36.5 |
| | **Moderate** | **128** | **43.2** |
| | Poor | 60 | 20.3 |
| Attitude towards PeKa B40 | Good | 54 | 18.2 |
| | **Moderate** | **184** | **62.2** |
| | Poor | 58 | 19.6 |
| Practice issues in PeKa B40 | **Good** | **192** | **64.9** |
| | Moderate | 94 | 31.8 |
| | Poor | 10 | 3.4 |
| Acceptance and level of participation in the NHFS | Good | 92 | 31.1 |
| | **Moderate** | **118** | **39.9** |
| | Poor | 86 | 29.0 |

**Table 4. Mean comparison of knowledge components (N = 296).**

| Knowledge components | Mean (SD) |
|---|---|
| Knowledge of health financing scheme | 3.77 (0.47) |
| Knowledge of gatekeeper roles | 4.08 (0.49) |
| Knowledge of PeKa B40 | 3.73 (0.62) |

**Table 5. Mean comparison of attitude components (N = 296).**

| Attitude components | Mean (SD) |
|---|---|
| Attitude toward health financing schemes | 3.76 (0.47) |
| Attitude toward gatekeeper roles | 3.96 (0.70) |
| Attitude towards PeKa B40 | 3.73 (0.52) |

**Table 6. Mean comparison of practice issues in PeKa B40 (N = 296).**

| Practice components | Mean (SD) |
|---|---|
| History | 4.50 (0.60) |
| Physical examination | 4.37 (0.54) |
| Investigation | 4.44 (0.62) |

**Table 7. Overall acceptance and level of participation in the National Health Financing Scheme (NHFS) (N = 296).**

| Variable | Mean (SD) |
|---|---|
| Acceptance and level of participation in the NHFS | 3.73 (0.52) |

## B. Bivariate analysis

As shown in Tables 8 and 9 below, simple linear regression and independent t-tests were performed to determine independent factors associated with the acceptance and level of participation of Malaysian GPs toward future NHFS. The significant associated factors were knowledge of health financing schemes, knowledge of gatekeeper roles, knowledge of PeKa B40, attitude towards health financing schemes, attitude towards gatekeeper roles, attitude towards PeKa B40, practice issues in Peka B40, gender, Family Medicine postgraduate qualification, and PeKa B40 provider status.

## C. Multivariate analysis

After adjustment for sociodemographic factors and other variables in multiple linear regression analysis, only PeKa B40 provider status, attitude towards health financing schemes,

**Table 8. Factors associated with the acceptance and level of participation of Malaysian GPs towards future NHFS (Simple linear regression).**

| Variables | b | 95% CI | P value |
|---|---|---|---|
| Age | 0.03 | -0.01, 0.07 | 0.138 |
| Knowledge of health financing schemes | 0.51 | 0.40, 0.63 | <0.001* |
| Knowledge of gatekeeper roles | 4.24 | 3.28, 5.19 | <0.001* |
| Knowledge of PeKa B40 | 2.12 | 1.32, 2.92 | <0.001* |
| Attitude toward health financing schemes | 5.02 | 4.06, 5.98 | <0.001* |
| Attitude toward gatekeeper roles | 3.71 | 3.10, 4.33 | <0.001* |
| Attitude towards PeKa B40 | 4.10 | 3.22, 4.98 | <0.001* |
| Practice issues in PeKa B40 | 2.98 | 2.05, 3.91 | <0.001* |

**Table 9. Factors associated with the acceptance and level of participation Malaysian GPs in future NHFS (Independent t-test).**

| Variables | Mean (sd) | t (df) | P value | 95% CI |
|---|---|---|---|---|
| **Gender** | | | | |
| Male | 22.89 (4.82) | 2.11 (292.14) | 0.036* | 0.07, 2.12 |
| Female | 21.79 (4.14) | | | |
| **Race** | | | | |
| Malay | 22.15 (4.71) | 1.09 (294) | 0.273 | -0.46, 1.63 |
| Others | 22.74 (4.34) | | | |
| **Nature of practice** | | | | |
| Solo | 22.12 (4.55) | 1.38 (294) | 0.167 | -0.32, 1.83 |
| Group | 22.88 (4.54) | | | |
| **Practice hours** | | | | |
| 24 hours | 22.62 (3.65) | -0.27 (294) | 0.789 | -1.99, 1.52 |

**Table 10. Factors associated with the acceptance and level of participation of Malaysian GP towards future NHFS using multiple linear regression (stepwise method alpha level 0.05).**

| | Adj. B | t stats | p-value | 95% CI | |
|---|---|---|---|---|---|
| **Constant** | -1.120 | -0.624 | 0.533 | -4.650 | 2.411 |
| **PeKa B40 provider** | | | | | |
| Yes | 0.960 | 2.265 | 0.024 | 0.126 | 1.795 |
| **Attitude toward health financing schemes** | 2.308 | 4.178 | <0.001 | 1.221 | 3.395 |
| **Attitude toward gatekeeper roles** | 2.383 | 6.775 | <0.001 | 1.691 | 3.075 |
| **Attitude towards PeKa B40** | 1.348 | 2.782 | 0.006 | 0.395 | 2.302 |

Adjusted R squared = 41.3%; The model fits reasonably well

attitude towards gatekeeper roles, and attitude towards PeKa B40 were associated with the acceptance and level of participation of Malaysian GP participation towards future NHFS ($p < 0.05$). As shown in Table 10 below, PeKa B40 providers had significantly higher acceptance and level of participation scores and were more prepared as compared to non-PeKa B40 providers by 1 score (p = 0.024). There was a significant linear relationship between attitude towards health financing schemes, gatekeeper roles, and PeKa B40 towards acceptance and level of participation towards future NHFS score at $p < 0.001$, $p < 0.001$, and p = 0.006 respectively. Those GP with 1 higher score in attitude towards health financing schemes have higher acceptance and level of participation score by 2 (95% CI: 1.221, 3.395). GP with 1 higher score in attitude towards gatekeeper roles have higher acceptance and level of participation scores by 2 (95% CI: 1.691, 3.075), and an increase of attitude towards PeKa B40 score increases acceptance and level of participation score by 1 (95% CI: 0.395, 2.302). R squared was 41.3%. The model fitted reasonably well, where model assumptions were met. There were no interactions between independent variables and no multicollinearity problems.

## Discussion

From the results, it has been observed that only four factors, which are namely PeKa B40 provider status, attitude towards health financing schemes, attitude towards gatekeeper roles, and attitude towards PeKa B40, are associated with the acceptance and level of participation of Malaysian GPs towards future NHFS. These suggest that attitude factors are more important determinants than knowledge factors in determining the acceptance and level of participation of the GPs in the NHFS. Additionally, this also implies that the GPs already possess a positive attitude and willingness on such matters and there will be less resistance on their part to participate in the NHFS. The following discussion focus on each area of study in greater depth.

### a. Knowledge

By Bloom's scoring classification, good knowledge of the health finance system, gatekeeper roles, and PeKa B40 are noted at 17.6%, 44.6%, and 27.0%, respectively. Malaysian GPs are very knowledgeable about their gatekeeper roles as compared to their levels of knowledge on health financing schemes and PeKa B40. This implies that the majority is already aware of this role they should be playing, although the policy has yet to be enforced. The GPs will be better prepared to assume this role without much difficulty. However, Malaysian GPs will need to have a more in-depth understanding of the types of health financing schemes that constitute their greatest deficit. This can be overcome by providing more information and education, namely through the internet, such as through the organisation of webinars and other

interactive courses, as these channels are the most popular sources of information, apart from the mass and social media.

The emphasis on lack of knowledge of health financing schemes has been similarly observed in a Kenyan study by Mathauer et al in 2008. 19 focus-group discussions of various groups in the informal sector have been done with the objective to analyse and understand the demand for social health insurance by informal sector workers in Kenya. The perceptions and knowledge of and concerns regarding social health insurance have been assessed. The conclusion is that the lack of knowledge about the NHIF (Kenyan National Hospital Insurance Fund) and its enrolment processes, has formed the most critical barrier [33].

### b. Attitude

Bloom's scores indicating good attitude is seen most strongly towards gatekeeper roles at 36.5%, followed by PeKa B40 at 18.2% and health finance schemes at 16.2%. The high scores on attitude towards gatekeeper roles synergises with their relatively high knowledge scores on the same area. The majority of respondents agree that GPs shall assume the full gatekeeper function. This indicates their strong willingness and readiness to execute this duty.

In a recent Indonesian study however, the scores on attitude towards the gatekeeper role by Indonesian GPs has attained lower values than knowledge. And the low level of attitude has influenced their performance [34]. This is in contrast to the findings of our study, in which the attitude component is more significant with higher scores than the knowledge factor.

### c. Practice

Practice matters relate to the various consultation processes in PeKa B40, which are the history, physical examination and investigation areas and each of their components. Overall Blooms' scores of 64.9% in the good category indicates a high acceptance rate to the inclusion of these processes.

### d. PeKa B40 provider status

Although the Peka B40 program is a good initiative of the government to initiate the public-private primary care partnership involving Malaysian GPs and the various KKs, the responses by the GPs in the survey are not forthcoming. A total of 60.5% (n = 179) of respondents are non-PeKa B40 providers.

### e. Acceptance and level of participation in the national health finance scheme

Bloom's scores indicate 31.1%, 39.9% and 29% under "good, moderate and poor" categories. The high moderate scores may partly reflect the lack of knowledge in health financing schemes and the low acceptance of the PeKa B40 scheme. Effective measures to resolve these issues may be able to result in a higher migration from the moderate to the good categories. However, a combined score of 71% for the good and moderate categories currently indicates a significant acceptance and participation level in the NHFS.

### f. Gatekeeper

As the first point of contact in the community. the GP needs to understand that the gatekeeper role requires a delicate balance between giving the best of medical care for his patient as opposed to the cost control measures. His professional competency as a primary care physician

managing his patient may reach a point where referrals for secondary and tertiary specialty care will be necessary over and above the issue of trying to reduce healthcare costs. That important judgemental skill has to be diligently observed whilst maintaining the patient's trust.

The fact that gatekeeping is already being practised in the government clinics in Malaysia, is all the more justification that GPs must be able to do the same. This will help to reduce inequalities and expand integrated care.

Hence, gatekeeping needed also to be introduced into the private sector. With regard to integrated care, the benefits of the gatekeeping function will also spill over to other allied health care givers. To begin with, this includes specific collaborative models in chronic disease management with community pharmacists. GP-led integrated teams are the way forward for primary care delivery.

## g. National health financing scheme

Recent events in Malaysia have given a very strong indication on the Malaysian government's commitment to transform the healthcare system and implement a national health financing scheme (NHFS). The 1Care system is the last comprehensive proposal that has a hybrid structure of a tax-based and social health insurance schemes. The reform will significantly result in a better coverage and positive outcomes which will be observable in the current situation. Unfortunately, the government of the day has to give in to opposition voices objecting to the social health insurance component. Worse still, one of the strongest oppositions come from the medical professionals, including the GPs and other secondary and tertiary clinicians, albeit for their own different reasons. This may be attributable to the lack of knowledge of social health insurance and the misconception of the gatekeeper roles amongst the doctors. Political rivalries further aggravate the issue.

However, the growing bipartisan movement to rejuvenate the NHFS recently, is a positive sign for Malaysia. But in the move to achieve this, it must be borne in mind that universal health coverage requires an effective structuring and management of health financing. Of utmost importance is a need for good governance and leadership so as to reduce inefficiencies and wastages of the various funds and resources. Without doubt, a strong primary care is required to achieving universal health coverage. The duality of public-private healthcare services in Malaysia needs to be well integrated at all levels of care.

One of the most relevant and successful models for the country to emulate comes from its nearest neighbour, that is Thailand. The country has ensured complete UHC for all segments of the population with a hybrid tax-based and social health insurance schemes. Additionally, the purchaser-provider split and gatekeeping at the primary care level, together with a comprehensive benefits package, has resulted in a highly successful structure. Underlying this is the fact that the support and collaboration between their healthcare professionals, civil societies and politicians has been crucial in ensuring its success.

This research extensively covers all 14 states in Malaysia, including Sabah and Sarawak. Additionally, it has been able to reach out to GPs in both urban and rural locations of each state in equal proportions. With a good response rate, the results of the study can be considered significant and representative of Malaysian GPs. The demographics are comprehensive and reflective of all matters that relate to general practitioners.

With the recent emphasis that the Malaysian government has on the implementation of an NHFS in the near future, the capacities of Malaysian general practitioners must be assessed in terms of their professionalism, knowledge, attitude, and skills in order to ensure that they will be able to contribute effectively.

The strength of this survey is that it is the first nationwide study that has involved Malaysian GPs both from the Peninsula and the states of Sabah and Sarawak. However, one limitation is the relatively low number of practising GPs in both urban and rural levels in the Federal territories of Labuan and Putrajaya, as well as the northernmost small state of Perlis. In that respect, it is not possible to perform a proper stratified random sampling and hence the only choice is to accept all the responses submitted.

**Theoretical contributions.**   As this is an original nationwide study and the model is proven to fit reasonably well as confirmed by the R squared value, the questionnaires in the survey and other aspects of the study design can be a platform for future studies with added refinements. Due to the scarcity of research materials on issues concerning private Malaysian GPs, this study will be able to stimulate more of such activities.

**Practical contributions.**   The results of this study should be strongly considered by the government in formalising the policy in defence of the interests of private Malaysian GPs in the NHFS. Firstly, in order to increase GP acceptance of the PeKa B40 programme, steps must be taken to allow GPs to fully manage their patients, with adequate remuneration and a more simplified system. Further, more aggressive promotional and informational efforts on the PeKa B40 system by ProtectHealth have to be carried out for both the GP community and the general public. The success of this initial phase of public-private partnerships will pave the way forward for more constructive partnerships. Secondly, the gaps in knowledge especially on health financing matters, need to be resolved. The GPs need to undergo related educational activities such as workshops and webinars. The role of GPs as gatekeepers must be implemented when NHFS is instituted. The government must acknowledge the importance of GPs in the primary care delivery and to encourage the development of vocationally-trained family physician-led integrated primary care teams for a cost-effective outcome. As important stakeholders, the GPs must be represented in policy making decisions.

**Future prospects of study.**   There are many more issues regarding private Malaysian GPs that require further research. These include public-private partnerships between government primary care doctors in community clinics (Klinik Kesihatan or KKs) and private GPs, case-mix and costings in primary care as well healthcare delivery by family physician-led multidisciplinary primary care teams.

## Conclusions

Malaysian GPs are largely positive in their attitude and participation in the forthcoming national health finance scheme, but less responsive towards the PeKa B40 scheme. The results of this study should be strongly considered by the government in formalising the policy in defence of the interests of Malaysian private GPs in the national health financing scheme. Of priority is to implement the gatekeeper role and increase the involvement in PeKa B40 by the GPs.

## Supporting information

**S1 Table. Knowledge of health financing schemes (N = 296).** This table contains the various questionnaires and the responses for each as according to the 5-point Likert scale.
(PDF)

**S2 Table. Knowledge of gatekeeper roles (N = 296).** The contents of this table focus on the responses to each aspect of the gatekeeper role as according to the 5-point Likert scale.
(PDF)

**S3 Table. Knowledge of PeKa B40 (N = 296).** This table indicates the responses according to the 5-point Likert scale on various aspects of the PeKa B40 scheme.
(PDF)

**S4 Table. Sources of information (N = 296).** This table contains the responses according to the 5-point Likert scale, on the various sources of information for each category ie. health financing schemes, gatekeeper role and PeKa B40.
(PDF)

**S5 Table. Attitude towards health financing scheme.** This table assesses the responses according to the 5-point Likert scale in reference to the attitude towards remuneration, tax-based finance, social health insurance and private insurance.
(PDF)

**S6 Table. Attitude towards gatekeeper roles (N = 296).** This table list the responses according to the 5-point Likert scale, on the attitude towards various aspects of the gatekeeper roles.
(PDF)

**S7 Table. Attitude towards PeKa B40 (N = 296).** This table list the various responses according to the 5-point Likert scale on the attitude towards various aspects of the PeKa B40 scheme.
(PDF)

**S8 Table. Practice issues in PeKa B40 (N = 296).** This table list the responses according to the 5-point Likert scale on various aspects of the consultation process in PeKa B40, including history, physical examination and investigation.
(PDF)

**S9 Table. Acceptance and level of participation to the National Health Finance Scheme (NHFS) (N = 296).** This table list the responses according to the 5-point Likert scale on the various issues related to the NHFS. This was to assess acceptance and level of participation in the NHFS.
(PDF)

**S1 File. Ethics statement.**
(PDF)

**S2 File. Respondent information sheet.**
(PDF)

**S3 File. Informed consent form.**
(PDF)

## Author Contributions

**Conceptualization:** Aznida Firzah Abdul Aziz, Syed Mohamed Aljunid.

**Data curation:** Mohammad Husni Jamal, Azimatun Noor Aizuddin.

**Formal analysis:** Mohammad Husni Jamal.

**Funding acquisition:** Azimatun Noor Aizuddin.

**Investigation:** Mohammad Husni Jamal.

**Methodology:** Mohammad Husni Jamal, Aznida Firzah Abdul Aziz, Azimatun Noor Aizuddin, Syed Mohamed Aljunid.

**Project administration:** Azimatun Noor Aizuddin.

**Resources:** Mohammad Husni Jamal, Azimatun Noor Aizuddin.

**Software:** Mohammad Husni Jamal.

**Supervision:** Aznida Firzah Abdul Aziz, Azimatun Noor Aizuddin, Syed Mohamed Aljunid.

**Validation:** Aznida Firzah Abdul Aziz, Azimatun Noor Aizuddin, Syed Mohamed Aljunid.

**Visualization:** Mohammad Husni Jamal.

**Writing – original draft:** Mohammad Husni Jamal.

**Writing – review & editing:** Aznida Firzah Abdul Aziz, Azimatun Noor Aizuddin, Syed Mohamed Aljunid.

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
