## [Decision Letter · Decision Letter 0]

2 Jun 2023

PONE-D-23-12708Gatekeepers in Health Financing Scheme: Assessment of Knowledge, Attitude,

Practices and Participation of Malaysian private GPs in PeKa B40 SchemePLOS ONE

Dear Dr. Aizuddin,

Thank you for submitting your manuscript to PLOS ONE. After careful consideration, we feel that it has merit but does not fully meet PLOS ONE’s publication criteria as it currently stands. Therefore, we invite you to submit a revised version of the manuscript that addresses the points raised during the review process.

I have now received the feedback from the reviewers, and after careful consideration, I have decided to revise the manuscript based on their comments.

Firstly, I would like to commend you on the valuable research you have conducted on the knowledge, attitude, practices, and participation of Malaysian private General Practitioners (GPs) in the PeKa B40 Scheme. The topic is timely and relevant, addressing an important aspect of health financing in Malaysia. The reviewers appreciated the significance of your study and its potential contribution to the existing literature.

However, there are several points that require attention and revision before the manuscript can be considered suitable for publication. I will outline these points below:

We look forward to receiving your revised manuscript.

Kind regards,

Naeem Mubarak, PhD

Academic Editor

PLOS ONE

Journal Requirements:

"Yes. The study was funded by the Faculty of Medicine, Universiti Kebangsaan

Malaysia FF-2020-403 under supervisor's name Azimatun Noor Aizuddin for PhD

student's research."

5. Please amend either the title on the online submission form (via Edit Submission) or the title in the manuscript so that they are identical.

Additional Editor Comments:

I have now received the feedback from the reviewers, and after careful consideration, I have decided to revise the manuscript based on their comments.

Firstly, I would like to commend you on the valuable research you have conducted on the knowledge, attitude, practices, and participation of Malaysian private General Practitioners (GPs) in the PeKa B40 Scheme. The topic is timely and relevant, addressing an important aspect of health financing in Malaysia. The reviewers appreciated the significance of your study and its potential contribution to the existing literature.

However, there are several points that require attention and revision before the manuscript can be considered suitable for publication.

Reviewers' comments:

Reviewer's Responses to Questions

**Comments to the Author**

1. Is the manuscript technically sound, and do the data support the conclusions?

Reviewer #1: Yes

Reviewer #2: Yes

2. Has the statistical analysis been performed appropriately and rigorously? 

Reviewer #1: Yes

Reviewer #2: Yes

3. Have the authors made all data underlying the findings in their manuscript fully available?

Reviewer #1: Yes

Reviewer #2: Yes

4. Is the manuscript presented in an intelligible fashion and written in standard English?

Reviewer #1: Yes

Reviewer #2: Yes

5. Review Comments to the Author

Reviewer #1: This study focuses on assessing the GPs' knowledge and attitudes towards health financing systems, their understanding of the gatekeeper roles within the health financing system, and their participation in the PeKa B40 scheme. The researchers employed a cross-sectional research design and used self-designed questionnaires to collect data from a sample of GPs across urban and rural areas in all 14 Malaysian states.

However, a few concerns need to be addressed before a final decision can be reached:

The literature cited does not depict an exhaustive coverage of the topic in hand. You need to bring more Malaysian studies and discuss in the discussion section.

Sample Selection: How was the stratified random sampling technique implemented to ensure the sample of GPs represents the population adequately? Were there any limitations or biases in the sampling method that could impact the generalizability of the findings?

Questionnaire Validity and Reliability: How were the self-designed questionnaires developed, and were they validated and tested for reliability? Can the researchers provide details on the measures taken to ensure the accuracy and consistency of the questionnaire responses?

Response Rate and Non-Response Bias: Although the response rate of 81.7% is reasonable, it is important to consider potential non-response bias. Were there any significant differences between respondents and non-respondents that could introduce bias into the findings?

Knowledge Assessment: The article mentions that GPs displayed lower scores in knowledge levels. Could the authors elaborate on the specific areas where knowledge gaps were observed? Were these gaps adequately addressed in the discussion and recommendations section of the article?

PeKa B40 Scheme: While the study associates positive attitudes towards the PeKa B40 scheme with acceptance and participation in the NHFS, it would be helpful to have a deeper understanding of the scheme itself. Can the authors provide more information about the PeKa B40 scheme, its objectives, and the specific aspects that need improvement based on the study findings?

Generalizability: Given that the study focused on private GPs in Malaysia, to what extent can the findings be generalized to other healthcare contexts or different types of healthcare providers? Are there any limitations in the study design or sample selection that may affect the external validity of the results?

Practical Implications: The article suggests that the results of the study should be strongly considered by the government. Can the authors provide more specific recommendations for policymakers and stakeholders based on their findings? How feasible are these recommendations in the current healthcare system?

Reviewer #2: Thank you for providing the opportunity to review this study. The study has a good deal of merit for publication with few improvements. Following are the recommendations for the authors to consider:

In the background section, the references to Malaysian studies lack depth. Would have been better to cite references from more recent studies concerning national health financing schemes in Malaysia. How would you define the role of “gatekeeping” that aligns with the objectives of your study? The premise set for the role of the private GPs as “gatekeepers” in health financing schemes is insufficient as to how their contribution can better the primary health care system of Malaysia. To elaborate more on this you may cite the following Malaysian studies:

• PMCID: PMC8064717

• PMCID: PMC6510413

Instead of mentioning the development of various national health schemes, the focus would have relied on schemes tailoring specifically to the needs of private GPs to support the rationale of your study.

Please provide more insights into the PeKa B40 health financing scheme with a compelling rationale to support its inclusion in national health financing schemes. How it can facilitate private GPs in improving the primary healthcare structure of Malaysia? Since PeKa B40 is a pilot public-private partnership program, why the study incorporated its use by private GPs only?

In questionnaire development, the authors state: “The self-designed questionnaires involved the contents from the literature review” However, no references have been cited to provide details of the literature review. Please provide details for the development of the questionnaire (how many statements were added/removed/modified) in the idea generation phase.

In the demographics section for the nature of the practice of private GPs, please clarify what you mean by “group practices” here. With reference to the “ownership” of private GPs please clarify what you mean by Partner, Director, and Employee. Do these correspond to the private GPs in university-based settings or private clinics?

In the discussion section, please state the knowledge, attitude, practice, and participation aspects of your study separately and explicitly with further mentioning the areas of improvement in knowledge, attitude, and practice areas. Requires mentioning of comparisons of health financing schemes in other low-middle income countries. Please mention the implications and future prospects of your study

6. PLOS authors have the option to publish the peer review history of their article (what does this mean?). If published, this will include your full peer review and any attached files.

Reviewer #1: No

Reviewer #2: No

---

## [Author Response · Author response to Decision Letter 0]

7 Aug 2023

We have responded to the 2 Reviewers comments in a tabulated form for ease of checking.

We hope this is acceptable.

---

## [Decision Letter · Decision Letter 1]

25 Sep 2023

Gatekeepers in the Health Financing Scheme: Assessment of Knowledge, Attitude, Practices, and Participation of Malaysian Private General Practitioners in the PeKa B40 Scheme

PONE-D-23-12708R1

Dear Dr. Azimatun Noor Aizuddin,

We’re pleased to inform you that your manuscript has been judged scientifically suitable for publication and will be formally accepted for publication once it meets all outstanding technical requirements.

Kind regards,

Naeem Mubarak, PhD

Academic Editor

PLOS ONE

Additional Editor Comments (optional):

Please revise the manuscript as suggested by Reviewer 2 before the final publication.

Reviewers' comments:

Reviewer's Responses to Questions

**Comments to the Author**

1. If the authors have adequately addressed your comments raised in a previous round of review and you feel that this manuscript is now acceptable for publication, you may indicate that here to bypass the “Comments to the Author” section, enter your conflict of interest statement in the “Confidential to Editor” section, and submit your "Accept" recommendation.

Reviewer #1: All comments have been addressed

Reviewer #2: All comments have been addressed

2. Is the manuscript technically sound, and do the data support the conclusions?

Reviewer #1: Yes

Reviewer #2: Yes

3. Has the statistical analysis been performed appropriately and rigorously? 

Reviewer #1: Yes

Reviewer #2: Yes

4. Have the authors made all data underlying the findings in their manuscript fully available?

Reviewer #1: Yes

Reviewer #2: Yes

5. Is the manuscript presented in an intelligible fashion and written in standard English?

Reviewer #1: Yes

Reviewer #2: Yes

6. Review Comments to the Author

Reviewer #1: All the queries have been addressed completely and the article is ready for publishing. Good Luck to all the authors.

Reviewer #2: The references should be updated. Please substitute more recent references for References # 3, 13, 28, and 31. After these minor changes, I have no further concerns to mention.

7. PLOS authors have the option to publish the peer review history of their article (what does this mean?). If published, this will include your full peer review and any attached files.

Reviewer #1: No

Reviewer #2: No

---

## [Editor Report · Acceptance letter]

6 Oct 2023

PONE-D-23-12708R1 

Gatekeepers in the Health Financing Scheme: Assessment of Knowledge, Attitude, Practices, and Participation of Malaysian Private General Practitioners in the PeKa B40 Scheme 

Dear Dr. Aizuddin:

I'm pleased to inform you that your manuscript has been deemed suitable for publication in PLOS ONE. Congratulations! Your manuscript is now with our production department. 

Kind regards, 

on behalf of

Dr Naeem Mubarak 

Academic Editor

PLOS ONE